# Evaluation of the Quality of Welded Joints After Repair of Automotive Frame Rails

**DOI:** 10.3390/ma18163849

**Published:** 2025-08-16

**Authors:** Andrzej Augustynowicz, Mariusz Prażmowski, Wiktoria Wilczyńska, Mariusz Graba

**Affiliations:** 1Department of Vehicle and Machine Mechatronics, Faculty of Mechanical Engineering, Opole University of Technology, Prószkowska Street 76, 45-758 Opole, Poland; w.wilczynska@student.po.edu.pl (W.W.); m.graba@po.edu.pl (M.G.); 2Department of Manufacturing and Materials Engineering, Faculty of Mechanical Engineering, Opole University of Technology, Prószkowska Street 76, 45-758 Opole, Poland; m.prazmowski@po.edu.pl

**Keywords:** automotive body repair, joint quality, microstructure analysis, passive safety, structural integrity, non-destructive testing

## Abstract

Passenger cars have unibody constructions, which means that their collision damage often involves key structural components. Successful repair requires the selection of appropriate technology and adherence to quality standards, which directly affects the safety of the vehicle’s continued operation. A commonly used method is a system of replacing damaged components with new ones, while repair by molding and forming is also possible—provided the original structural features are preserved. Automotive body repairs require advanced welding techniques and high precision. Methods such as MIG, TIG, as well as brazing and soldering have replaced older techniques, providing more efficient joining of HSS and HSLA components. Maintaining quality workmanship is crucial, as repair errors can weaken a vehicle’s structure and compromise passenger safety. This article presents the results of a study on the evaluation of the quality, microstructure, and mechanical properties of welded joints of a passenger car frame rail section made of high-strength, low-alloy steel—HSLA 320. The joints were made by three welding methods: MMA, MAG, and TIG, using different technological parameters. Microstructural analysis, non-destructive testing, and microhardness measurements made it possible to assess the impact of the chosen technology on the quality and strength of the joints. The best results were obtained for the TIG method, characterized by the highest repeatability and precision.

## 1. Introduction

The automotive industry is being created by imperatives of environmental sustainability, energy efficiency, and increased productivity [1]. The automotive industry is being created by imperatives of environmental sustainability, energy efficiency, and increased productivity [2]. Advances in automotive technology reveal a trend toward producing faster, more powerful, and lighter vehicles. One of the critical criteria in modern car design is ensuring overall safety, particularly in terms of stability and vehicle control [3]. In designing the car body, it is essential to consider both cost and customer comfort. Today, car body manufacturing is becoming increasingly complex due to market competition. Customers evaluate vehicles not only based on efficiency but also on aesthetics and design. As global regulations tighten the fuel economy and emissions standards, automakers are being forced to seek innovative materials and design solutions to significantly reduce vehicle weight while maintaining or improving structural integrity and resistance to deformation [1]. Vehicle efficiency depends on both the internal combustion engine and the aerodynamic design of the body. Customers also expect vehicles to provide both active and passive safety features [4]. The automotive sector is under growing pressure to enhance design efficiency, reduce emissions, and implement sustainable technologies. As a result, key vehicle components are being redesigned. These components serve a dual purpose: they must meet visual and design expectations while ensuring passenger safety by absorbing lateral impact energy during collisions [5]. In response, manufacturers are redesigning crucial vehicle parts and introducing advanced production methods, particularly in the development of interior door panel systems [6]. Throughout a vehicle’s lifetime, various external factors act upon its body. These influences may alter the mechanical and geometric properties of structural elements. Changes in the mechanical characteristics of metal structural components can result from numerous continuous and random factors, both in terms of nature and intensity [7]. A separate category of external influences includes those caused by traffic accidents and subsequent repairs. According to traffic accident statistics, most collisions are frontal, which is why significant attention is paid to the front crumple zones during vehicle body design [8]. Vehicles are frequently repaired after accidents and continue to be used afterward. Repairs often involve restoring the vehicle body’s original geometric parameters. However, no standardized specifications currently govern the repair process. Post-collision vehicle repair entails significant technical and economic considerations. It is relatively common for pre-owned vehicles to exhibit a damage history associated with prior traffic collisions. Operational safety is governed primarily by technical regulations that may impose constraints on the vehicle’s continued serviceability. In frontal impact scenarios, for example, the longitudinal structural members often experience a loss of local stability due to axial compressive loads and contribute to energy dissipation through progressive plastic deformation and buckling mechanisms. The restoration of these components to their original structural integrity is a critical, labor-intensive procedure, requiring precise alignment, material evaluation, and often complex repair or replacement strategies to ensure compliance with safety and performance standards [8,9,10]. As the number of older vehicles continues to increase and a high number of defective vehicles continues to increase, it is crucial to understand the impact of vehicle defects on road safety and to identify strategies to mitigate the risk of these incidents. Vehicle defect-related crashes have been a persistent safety concern, with design flaws, manufacturing errors, and lack of maintenance leading to numerous incidents.

## 2. Selected Problems of Post-Accident Repair of Car Body

The bodies of modern passenger cars are designed and manufactured mainly as unibody constructions. Thus, damage to the body in a traffic incident often involves the deformation of important structural elements of the car. Repairing such damage requires the appropriate selection of repair technology and observance of workmanship quality. On this, the safety of further use of the car depends [11]. A common system for repairing a vehicle’s body is the assembly replacement system, which involves removing a faulty component in the repair process and installing in its place a new one or one obtained from a vehicle of the same make and type from recycling. Repairing structural components of a vehicle requires the use of quality replacement components provided by the vehicle manufacturer for use. In justified cases, molding and plastic processing technology is allowed when repairing the body. Special care must then be taken in reproducing the shape and design features of the repaired component [12].

When applying body repair using partial replacement, molding, and forming technologies, inaccuracy in reproducing the shape of the repaired component must be taken into account. There is also a risk of losing the strength and functional parameters of the repaired structure, causing a decrease in the level of passive safety of the vehicle. Particularly noteworthy is the technology of the partial replacement, during which the process of joining the replaced section by welding is performed. During this process, the material structure is changed, and the repaired part is weakened in the joining area. Such an example is, for example, the repair of front frame rails, the structure of which is the so-called controlled crush zone with the division of its deformation [11]. Unprofessional car repair can have a negative impact on the behavior of a vehicle involved in a subsequent accident. It is very important that the repair of such a car is carried out with the appropriate technique and quality of workmanship. An analysis of this topic has been carried out for years by the KTI (Kraftfahrzeugtechnisches Institut) in Lohfelden, determining in a study how an improperly performed car repair after an accident can affect the safety of passengers in the event of subsequent damage. The focus was mainly on the effects of vehicle side impacts, simulated during crash tests. It was shown during the crash test that a VW Passat B6 car that was not properly repaired after an accident significantly lost passive safety [13]. And damage to a properly repaired vehicle of the same make and type showed deformations comparable to those of a car in its original condition. These results led to the conclusion that adherence to the vehicle manufacturer’s specifications is a basic requirement for professional repairs. In the case at hand (for a VW Passat B6 car with a side impact according to ECE Regulation No. 951), passive safety was fully preserved after professional repair [14]. As part of the “Fair Repair” project of the Kraftfahrzeugtechnisches Institut (KTI), two identical Škoda Octavia cars that were damaged in a side impact were examined, based on previous tests. Both vehicles were subsequently repaired. One vehicle was repaired but not according to the manufacturer’s specifications; the other was repaired at an authorized Škoda service center according to the manufacturer’s specifications. The repaired vehicles were damaged again as a result of the side impact. Test results were compared in terms of the deformation behavior of the vehicles’ bodies and the dummies placed in the vehicles. Analysis of acceleration measurements revealed significantly higher passenger loads in the vehicle repaired incorrectly compared to the vehicle repaired according to the manufacturer’s specifications. As for the analysis of the extent of damage, significantly deeper deformations were revealed in the vehicle repaired incorrectly in the areas of welded joints [15,16]. Researchers at the University of Agder conducted two full-scale crash tests at Farsund Airport Lista in 2021. The findings show that vehicles improperly repaired after a collision are significantly less safe if involved in another crash. It was found that when a vehicle damaged in a collision is not repaired according to official standards, there is an increased risk of more extensive structural damage in the event of a subsequent major collision. As a result, such crashes may also lead to more serious injuries for the vehicle’s occupants [17]. Collision repair of a vehicle is a process defined by the automaker for each product in its portfolio. It includes detailed procedures for repairing or replacing parts of the vehicle. The repair manual is a comprehensive document that outlines the appropriate methods for each component, taking into account structural properties, the effects of heat treatment, and crashworthiness requirements. Most automakers recommend replacing structural components after plastic deformation and strictly prohibit heat-based repairs on the body and frame parts. For components made from ultra-high-strength steel (UHSS), reinforcement repairs are typically not advised in order to preserve the vehicle’s crashworthiness and ensure that occupant protection features remain intact. However, it has been observed in some cases that repair technicians do not follow the procedures specified by the automakers. This non-compliance can result in serious safety risks for vehicle occupants in the event of a crash [18].

## 3. Steel as a Structural Material for Automotive Parts

In the 1950s, the automotive industry began using substitutes for the structural material wood, replacing it with low-carbon steel, aluminum, and biopolymer [19]. Steel soon became the dominant material. But by the end of the last century, a lot of research had already been performed on replacing steel as the leading material in the body structure, and a number of vehicle prototypes using aluminum and polymer composites were proposed, resulting in significant reductions in vehicle weight [20]. Nowadays, various materials (steel, aluminum, magnesium, and polymer composites) produced in many forms (sheet, extruded, and cast) are used in the body structure [21] (Figure 1).

Automotive steel includes various types of steel products. Some of them are used to build the drivetrain and suspension. The other group of products is used for the body. For the car body, advanced high-strength steel (AHSS) is generally considered the best material. This is an alloy with additional metal elements and solutions to increase strength. The key feature of AHSS is its tensile strength [22]. AHSS grades use a combination of strengthening mechanisms based on strength and ductility requirements. From a historical perspective, AHSS can be divided into first-, second-, and third-generation AHSS [23]. The current and emerging steel grades are presented in Figure 2 [24]. First-generation AHSS has a multi-phase and complex microstructure. Nevertheless, traditional high-strength steel HSS with ferrite and martensite phases gave a higher strength and better formability than first-generation AHSS but did not reduce the weight enough. Some of the most commonly used grades in first-generation AHSS include dual-phase steels (DP), transformation-induced plasticity (TRIP) steels, press-hardened (PH) steels, and complex-phase (CP) steels. The second generation of AHSS used more alloying elements to stabilize austenite than the first generation, resulting in a higher strength and ductility. Steels with the effect of twinning-induced plasticity (TWIP), austenitic stainless steels (ASSs), and light steels with the addition of Al, also with induced plasticity (L-IP), mainly belong to the second generation of AHSS. Third-generation AHSSs, on the other hand, are materials with parameters between the first and second generations. Third-generation AHSSs include medium-manganese steels, quenched and partitioned (Q&P) steels, and carbide-free bainitic steels (CFB) [25,26].

### Welding Techniques Used in Vehicle Repairs

Welding performed during auto body repair is quite different and often more demanding than in many other welding professions. Welding ultra-thin, complex body panels requires a great deal of skill and knowledge. Current welding techniques and equipment are displacing the oxy-acetylene processes previously used. Gas metal arc welding (GMAW)—better known as metal electrode inert gas (MIG) welding—offers many advantages for joining high-strength steel (HSS) and high-strength, low-alloy steel (HSLA) components used in modern automobiles [12].

Car manufacturers recommend several types of welding from all positions (horizontal, vertical) during accident repairs. Body welders use the following welding techniques and, in justified cases, soldering (Figure 3) [12]:Gas metal arc welding (GMAW), sometimes referred to by its subtypes metal inert gas (MIG) and metal active gas (MAG), is a common welding process. The MIG method is the most common type of welding for steel one-piece panels, as well as medium and thick frames. This method is also used for aluminum panels but with thicker aluminum wire. The MAG method is used in industrial and structural applications, especially for carbon steels and low-alloy steels.Manual metal arc (MMA) welding, also known as shielded metal arc welding (SMAW) or stick welding, is a versatile welding process that uses a consumable electrode coated with flux to create an arc and join metals. The flux coating melts during welding, producing a shielding gas and slag to protect the welded area from atmospheric contamination. MMA welding is widely used in construction, shipbuilding, pipeline construction, and repair work due to its simplicity and ability to be used in various welding positions.TIG (Tungsten Inert Gas) is an arc welding process using a non-fusible tungsten electrode in an inert gas shield. The filler wire is selected according to the material to be welded. The method is often recommended when welding aluminum alloy body panels.Soft brazing is recommended for joining and sealing the corners of roof panels and other large area panels.Hard brazing is used in areas where strong and durable connections are needed that can withstand harsh working conditions such as high temperature, pressure, and vibration. Examples include repairs to the cooling system, fuel system, or engine components.

## 4. Object of Study

The purpose of the presented research is to evaluate the effect of selected welding methods (MMA, MAG, and TIG) and varied current parameters on the mechanical properties and structural changes in welded joints made on passenger car frame rail material. The subject of the study is a representative fragment of a real vehicle superstructure, in the form of part of a passenger car frame rail, made of high-strength, low-alloy steel (HSLA), designated as HSLA 320. This element was extracted for mechanical and structural testing. The analyzed section of the stringer, shown in Figure 4, has dimensions of 300 mm × 60 mm × 60 mm and was made of 0.8 mm thick sheet metal, typical for body components subjected to both static and dynamic loads. The dimensions and geometric parameters of the specimen were chosen to enable it to reflect actual operating conditions in the context of further strength and technological analyses.

The chemical composition of the steel used was determined using a MiniLab 300 spark spectrometer (G.N.R. S.r.I., Agrate Conturbia, Italy). Measurements were carried out five times, and the average values of the content of chemical elements were compared with the reference values in Table 1. The data obtained allow for the unambiguous identification of the material and the evaluation of its compliance with the standards for the HSLA 320 grade. Table 2 shows the mechanical properties according to the standard.

The structure of HSLA 320 steel, shown in Figure 5a,b, is fine-grained ferrite with a small amount of pearlite. Inside the ferrite grains there are micro-separations of carbides and nitrides of Nb, Ti, and V. This microstructure provides good strength, high impact toughness, formability, and high weldability.

In order to achieve the research objectives, six welded specimens (the schematic welding system of the samples is shown in Figure 6) were prepared using three different welding methods: manual arc welding with a covered electrode—MMA (Figure 7a,b), welding with an inert gas shielded electrode—TIG (Figure 8a,b), and arc welding with a consumable electrode shielded by active gases—MAG (Figure 9a,b). Two specimens were prepared for each of the aforementioned methods, using different technological parameters to evaluate the effect of process conditions on the quality and properties of the welded joint. The designations of the individual specimens, along with the detailed values of the welding parameters used during their preparation, are shown in Table 3.

## 5. Research Methodology

The tested elements in the form of welded frame rail sections were subjected to an evaluation of the quality of the joints made using non-destructive testing. Two diagnostic techniques were used: visual testing (VT—visual testing) and penetrant testing (PT—penetrant testing). Visual testing was carried out in accordance with current standards [28] to initially evaluate the welded joints for the presence of weld incompatibilities such as undercuts, excessive overfills, craters, weld face irregularity, and cracks. This was followed by penetrant testing to detect any inconsistencies not visible by visual inspection. Samples for further testing, structural and mechanical, were cut using the electrical discharge machining method (WEDM), implemented on an AgieCharmilles CUT E wire-cutting machine, manufactured by GF Machining Solutions (Bern, Switzerland). The use of this technology ensured high dimensional accuracy and minimal thermal impact on the material structure. In order to evaluate changes in material strengthening and to analyze the microstructure of the welded joints, metallographic specimens were cut, including the weld (W), heat-affected zone (HAZ), and base material (BM) areas. The cut specimens were encapsulated in a thermosetting non-phenolic resin and then subjected to manual grinding using abrasive papers with gradations from #150 to #2000. Final polishing was carried out on polishing cloths using an aqueous alumina suspension (Al_2_O_3_), which made it possible to obtain the specular surface necessary for further analysis. Hardness tests were carried out using the Vickers method with a load of 1 kg (HV1), using a Micro-Vickers SMV-2000MZ hardness tester (SUNPOC, Guiyang, China). For each specimen, 45 impressions were made, spaced at equal intervals of 0.5 mm along a line drawn through the center of the specimen’s cross-section (Figure 10).

In parallel, microstructural studies were carried out using an OLYMPUS IX70 optical microscope (OLYMPUS Corporation, Tokio, Japan) equipped with a digital camera by OPTA-TECH (Warsaw, Poland) together with dedicated image analysis software designed for this camera. In order to visualize the grain structure and phase boundaries, the samples were etched in a 5% solution of nitric acid in ethanol (nital) after polishing. In order to determine the mechanical properties of the welded joints, five static tensile test specimens were cut from the prepared parts for each joint variant. The shape and dimensions of the specimens are shown in Figure 11. Tensile tests were carried out using an Instron Electropuls 10000 hydraulic testing machine (Instron Worldwide, Norwood, Massachusetts, USA), enabling the controlled measurement of force and elongation until the specimen broke. The obtained results were used to evaluate the tensile strength of the tested joints.

## 6. Research Results and Their Analysis

Macroscopic and microscopic metallographic analyses are performed to assess the structure of a metallic material, which allows for the determination of its properties, quality, and potential defects. The macroscopic evaluation of the metal’s structure is carried out with the naked eye or at a low magnification (typically up to 20×). It reveals inhomogeneities such as cracks, non-metallic inclusions, porosity, delamination, chemical segregation, and the structure of welded joints. Microscopic evaluation is a detailed assessment of the metal’s microstructure at high magnifications (typically 50×–1000×). It reveals structural components such as grains, grain boundaries, metallic phases, precipitates, and products of phase transformations.

### 6.1. Macro and Microscopic Metallographic Analyses

Macroscopic analysis revealed the occurrence of welding inconsistencies of the following types: irregular width and unevenness of the weld face (Figure 8a and Figure 9a), concave face (Figure 8a and Figure 9b), improper welding start—craters (Figure 7a,b and Figure 9b), excessive convexity of the face (Figure 12a), linear misalignment (Figure 12b,f), porosity (Figure 12c), and lack of fusion (Figure 12c). Figure 12e presents sample 2, in which no welding defects were detected. A detailed macroscopic examination using the penetration method (PT) did not reveal the presence of additional inconsistencies, only the presence of craters was confirmed. Example results of non-destructive testing of an MMA-welded sample are shown in Figure 13a,b.

Measurements made for the specimens shown in Figure 7, Figure 8 and Figure 9 made it possible to determine the width of the heat-affected zone (HAZ) formed during welding by various methods. The narrowest HAZ of about 30.5 mm was observed for specimen 5, made using the TIG method with a welding current of 50 A. In contrast, the widest zone, about 47 mm, was obtained for specimens No. 2 and 4, made using the MMA method (60 A welding current) and the MAG method (55 A current), respectively. In the case of sample No. 4, despite the use of lower welding parameters than in specimen 3 (MAG with a current of 95 A), the greater impact of HAZ is probably due to the execution of two beads in sample No. 4, which resulted in the introduction of more heat into the material.

The impact of high temperature significantly affects structural changes in welded materials. Evaluation of these changes was made possible by microscopic observations. The structure of the weld (FZ), the heat-affected zone (HAZ), and the base material (BM), using the example of the joint in sample No. 4 (Figure 14a), is shown in Figure 14g–j.

In the welding process of HSLA 320 steel, as a result of high-temperature phase transformations, a ferritic–bainitic structure was identified within the weld (FZ) (Figure 14g). It forms as a result of complete melting of the material in this region, followed by solidification from the liquid phase through crystallization and subsequent cooling. Bainite forms as a product of the austenite transformation at moderately high cooling rates. In the heat-affected zone (HAZ), depending on the distance from the fusion boundary, subzones with a ferritic–bainitic structure (Figure 14h) and a ferritic–pearlitic structure (Figure 14i) were distinguished. The HAZ (heat-affected zone) is a region that has not undergone melting but has been significantly overheated, leading to microstructural changes. Different subzones can be distinguished depending on the distance from the fusion line and the peak temperature reached. Closer to the FZ, the microstructure is predominantly ferritic–bainitic (Figure 14h), similar to the weld metal but more irregular. It forms as a result of the rapid cooling of austenite. Further from the FZ, a ferritic–pearlitic structure develops (Figure 14i), typical for lower thermal exposure temperatures. For HSLA-type steels, the formation of needle-like ferrite (acicular ferrite), characterized by a lamellar morphology, is characteristic of the result of phase transformations during cooling. In addition to HAZ, the structure of the base material (BM), that is, ferrite with a small amount of pearlite, was identified (Figure 14j).

The structures of samples 1–3 and 5–6 are shown in Figure 14b–f.

### 6.2. Mechanical Testing

The next stage of the research involved analyzing the influence of current parameters in welded joints made using three methods, MMA, MAG, and TIG, on the hardness distribution in the joint. Subsequently, the results of the static tensile test of the welded joints were compared.

#### 6.2.1. Analysis of Hardness Distribution

Hardness measurements were made using the Vickers (HV) method in three characteristic areas: the base material (BM), the heat-affected zone (HAZ), and in the weld (FZ). The purpose of the analysis was to evaluate the effect of the thermal parameters of the process on the mechanical properties of welded joints. In the case of MMA-welded specimens, despite the use of a lower welding current (40 A) in specimen 1 compared to 60 A in specimen 2, the first weld has a significantly higher average hardness (223 HV and 201 HV, respectively) and a wider range of variation (208 to 244 HV and 188 to 229 HV), as shown in Figure 15. This indicates that the lower current in sample 1 resulted in faster cooling and thus in higher hardness structures, such as bainitic. In contrast, the higher welding current used in specimen 2 introduced more heat into the joint, resulting in slower cooling and the formation of a more homogeneous but less hard structure, probably with a higher proportion of ferrite. These differences confirm the significant influence of the thermal parameters of the welding process on the structural characteristics and mechanical properties of the welds.

For the MAG method, the hardness distribution was analyzed for specimen 3 (90 A) and specimen 4 (55 A), as shown in Figure 16. The higher current in specimen 3 resulted in a pronounced increase in hardness in the HAZ (242 HV for pr. 3 and 211 HV in specimen 4), which can be attributed to more intense thermal interactions and hardening transformations in this zone. The hardness of the welds was similar (235 HV in specimen 3 and 231 HV in specimen 4), while the base material retained comparable properties in both samples (179 to 182 HV), indicating its stability. An analogous relationship was observed in Figure 17 in the TIG method (specimens 5 and 6). Higher currents (76 A in sample 6) resulted in an increase in HAZ hardness (228 HV) in comparison to specimen 5 (50 A, average 211 HV). An inverse relationship was found in the weld, where the lower current in specimen 5 resulted in an increase in hardness (253 HV) relative to specimen 6 (237 HV), which is related to faster cooling with less heat input. The base material in both cases retained an almost identical hardness (~178.5 HV), confirming the homogeneity of the base material.

The presented test results showed a significant effect of current parameters on the local hardness distribution in welded joints. For all welding methods, an increased HAZ hardness is observed with increasing current intensity and an increase in weld hardness at lower current values, which is due to the different cooling conditions and accompanying phase transformations. The base material remained unchanged in hardness, confirming that the main mechanical differences are concentrated in the heat-affected zones and in the weld itself.

#### 6.2.2. Static Tensile Test

The comparison of the static tensile test results for the welded joints is presented in Figure 18. Based on the results obtained from five repetitions, the average tensile strength was calculated. The base material, whose average tensile strength R_m_ was 489 MPa, was used as a reference. For MMA-welded joints, average R_m_ values of 540 MPa (specimen 1) and 544 MPa (specimen 2) were obtained. Both values exceed the strength of the base material before the welding process, which indicates the good quality of the fabricated joints. The difference between the samples was small, which may indicate the relatively low sensitivity of this method to changes in current parameters within the tested range. The joints made using the MAG method achieved average values of R_m_ = 537 MPa (specimen 3) and R_m_ = 545 MPa (specimen 4). Both results also exceed the strength of the base material, but there is a noticeably greater difference between the samples compared to the MMA method. This may suggest a greater dependence of the MAG joint quality on the precise selection of welding parameters, particularly the welding current. The highest uniform values of tensile strength (R_m_ = 548 MPa) were obtained for TIG-welded specimens (specimen 5 and 6). The obtained repeatability and the highest R_m_ values among all the results testify to the high precision of this method and its low sensitivity to the change in current intensity in the analyzed range. TIG provided the most homogeneous and robust joints among all the methods tested.

All the welding methods analyzed produced welded joints with a higher strength than the native material. The best results were obtained for the TIG method, which also proved to be the most repeatable. The MMA and MAG methods showed similar results, but MAG was more susceptible to changes in welding parameters, which may be important when selecting technology for specific automotive repair applications.

## 7. Discussion

In automotive structures, longitudinal beams play a key role as load-bearing elements, transferring dynamic loads and ensuring the rigidity of the chassis. Consequently, welded joints in these components must exhibit high static strength, fatigue resistance, and microstructural homogeneity. The research conducted in this study on welded joints of HSLA 320 steel, produced using MMA, MAG, and TIG methods, enables the comparison of the results with conclusions presented in the available literature.

Macroscopic analysis confirmed the presence of typical welding discontinuities, such as crater cracks, lack of fusion, porosity, and excessive weld reinforcement, particularly in joints made using the MMA and MAG methods. Such defects are described in the literature [29]. Porosity typically results from gaseous contaminants, moisture, or uncontrolled cooling, while a lack of fusion may be caused by an insufficient welding current or swift electrode movement. Crater cracks form due to improper weld termination, and excessive reinforcement is typically due to excessive heat or filler material input. Methods such as TIG and laser welding significantly reduce the occurrence of these defects due to greater thermal precision and process stability [30].

Microstructural analysis confirmed a clear differentiation of phases within the fusion zone (FZ), heat-affected zone (HAZ), and base material (BM). In particular, ferritic–bainitic and ferritic–pearlitic structures were identified, with their distribution strongly dependent on the welding process parameters, primarily the current intensity. These findings are consistent with the observations of various authors [31,32] for the welded joints of S960MC and DP600 steels welded by laser, where bainitic and martensitic structures dominated in the FZ. At the same time, the HAZ showed a mixture of bainite, ferrite, and martensite. The proportions of these phases varied depending on the distance from the fusion line and the cooling rate. Furthermore, in microalloyed low-carbon steels containing Mo, V, Ti, N, and B [33], it was shown that an increase in linear heat input caused a transformation in the CGHAZ from bainitic and martensitic to acicular ferrite and polygonal ferrite. These transformations resulted from reduced cooling rates and prolonged thermal cycles.

The Vickers hardness tests conducted in this study revealed that lower welding currents led to increased hardness within the weld metal and a wider range of variability due to more intensive phase transformations. Conversely, higher currents resulted in more homogeneous but less complex structures. This relationship aligns with findings in the literature [30], where a lower heat input or welding current favors hardness increases due to bainitic structure formation [34], while higher values lead to reduced hardness due to coarse-grained microstructures.

Tensile tests of the joints showed that all welding methods, MMA, MAG, and TIG, enabled higher tensile strength values than the base material, indicating high-quality welds. Notably, the TIG method stood out due to its superior repeatability and maximum strength values, as confirmed by the literature [30]. TIG welds are often characterized by the best mechanical properties [29] and offer significantly more stable joints due to precise heat input control than MIG/MAG or MMA processes [30].

All the aforementioned findings are consistent with the results obtained for HSLA 320 steel welded joints and confirm the crucial role of thermal parameters, particularly the welding current, in shaping the mechanical and structural properties of load-bearing elements used in the automotive industry. Especially for longitudinal beams, where reliability is critical for safety, welding methods with a high thermal precision (e.g., TIG) should be preferred, as they ensure homogeneous microstructures, minimize defects, and provide stable mechanical properties, even with slight variations in process parameters.

As demonstrated by the presented results and referenced studies, properly selected welding parameters and appropriate technological procedures can enable the localized repair and restoration of satisfactory mechanical properties of these materials. However, the welding process causes localized exposure to high temperatures, which leads to microstructural changes in the heat-affected zone (HAZ), such as recrystallisation, grain growth, phase transformations, and the loss of dispersion strengthening effects. Quenched or dual-phase steels (e.g., DP600, CP800, and martensitic UHSS) result in the potential degradation of mechanical properties, including a reduced yield strength and fatigue resistance. For this reason, manufacturers discourage any attempts to segment or weld such structural components after accidents [18]. In the event of damage or cracking, following official repair procedures by OEM documentation and international safety standards is imperative.

## 8. Conclusions

The effect of three different welding methods—MMA, MAG, and TIG—on the microstructure, hardness, and strength of the joints was analyzed, while taking into account the occurrence of welding inconsistencies and the variability of technological parameters. The results obtained made it possible to assess the suitability of each method in the context of repair requirements of vehicle structural components. Based on the study, a number of conclusions can be made about the quality, structure, and mechanical properties of welded joints made using HSLA 320 steel.

HSLA 320 steel shows very good mechanical and structural properties, predisposing it to applications in vehicle load-bearing components.The quality of welded joints largely depends on the method used and the technological parameters. The lowest number of nonconformities was obtained for the TIG method.Welding parameters, especially current intensity, have a significant effect on the hardness distribution and microstructure of joints. Lower currents favor the formation of harder and finer-grained structures.The impact of the heat-affected zone (HAZ) was greatest when welding with MMA and MAG methods at higher current parameters or using multi-pass weld joints.All welding methods made it possible to produce joints with tensile strengths higher than the native material, confirming their suitability for structural applications requiring high-quality joints.The TIG method has proven to be the most effective, providing the highest strength values and the greatest repeatability of results, making it particularly advantageous in applications requiring precision and uniformity of connections.MAG welding showed a greater sensitivity to changes in current parameters than MMA and TIG, indicating the need for careful selection of settings to achieve optimal joint quality.The MMA method, despite its simplicity, has provided stable and good quality joints, which may give an argument for its use where process robustness for small parameter variations is important.The results obtained can provide a basis for optimizing welding technology according to strength requirements and production conditions, especially in the structural and engineering industries.

## Figures and Tables

**Figure 1 materials-18-03849-f001:**
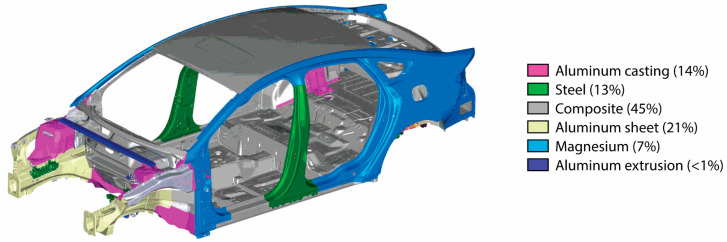
Mixed-material lightweight vehicle Mach II body-in-white material distribution [21].

**Figure 2 materials-18-03849-f002:**
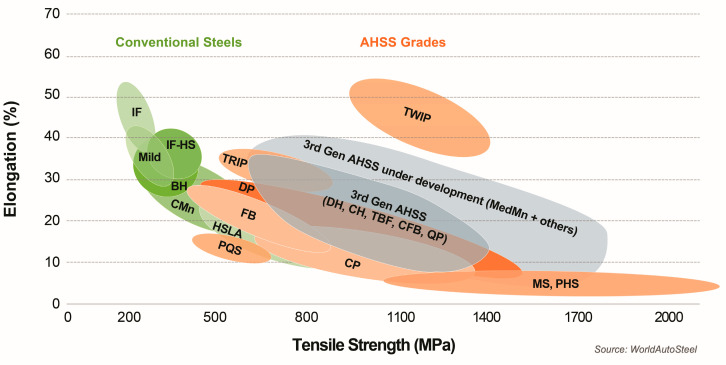
The Global Formability Diagram (2021) comparing strength and elongation of current and emerging steel grades (Courtesy of WorldAutoSteel) [24].

**Figure 3 materials-18-03849-f003:**
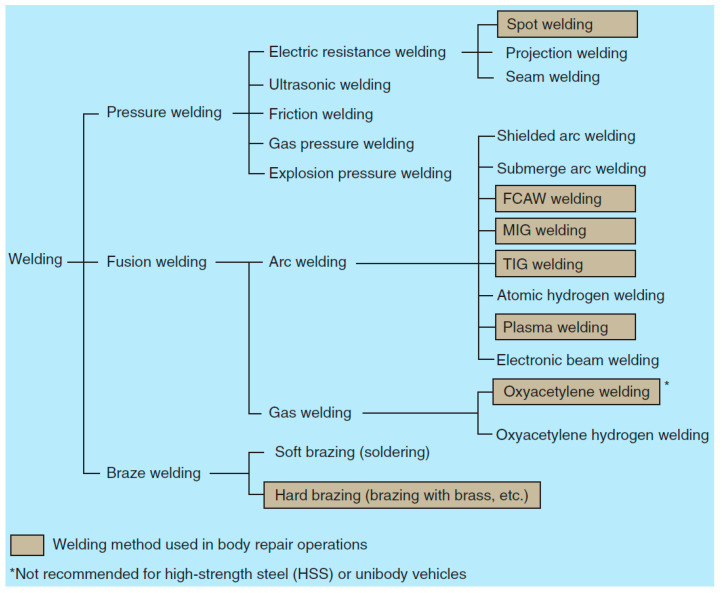
Welding methods [12].

**Figure 4 materials-18-03849-f004:**
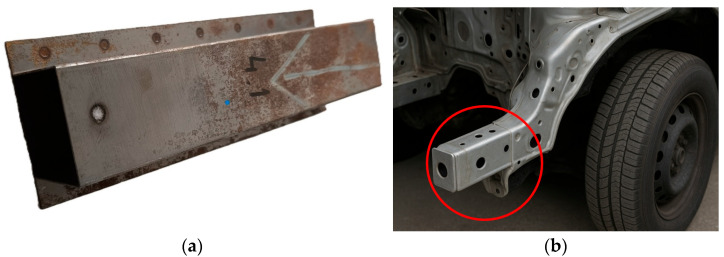
The view of an automotive frame rail made of steel: (**a**) the subject of the study, (**b**) part located on the vehicle.

**Figure 5 materials-18-03849-f005:**
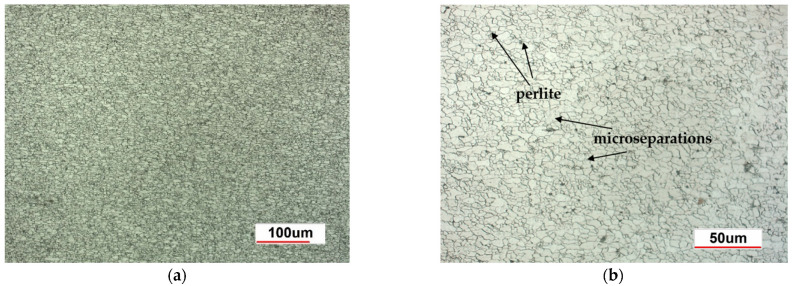
Structure of HSLA 320 steel—ferrite with small separations of pearlite: (**a**) magnification 200×, (**b**) magnification 500×.

**Figure 6 materials-18-03849-f006:**
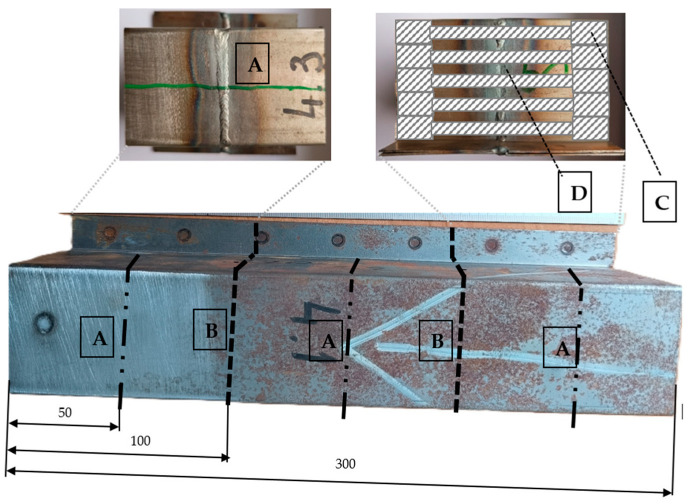
Welding scheme of the samples: (A) cutting line of the longitudinal beam along which the weld was made, (B) cutting line of the sample for the selected welding method, (C) area of sample extraction for the static tensile test, and (D) area of sample extraction for metallographic examination and hardness measurement.

**Figure 7 materials-18-03849-f007:**
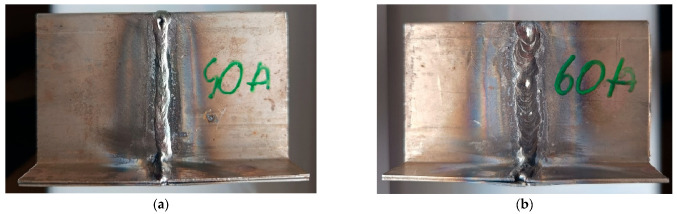
MMA welded parts: (**a**) specimen 1—welding current 40 A, (**b**) specimen 2—welding current 60 A.

**Figure 8 materials-18-03849-f008:**
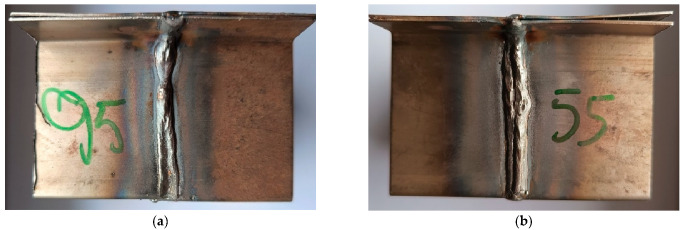
MAG welded parts: (**a**) specimen 3—welding current 95 A, (**b**) specimen 4—welding current 55 A.

**Figure 9 materials-18-03849-f009:**
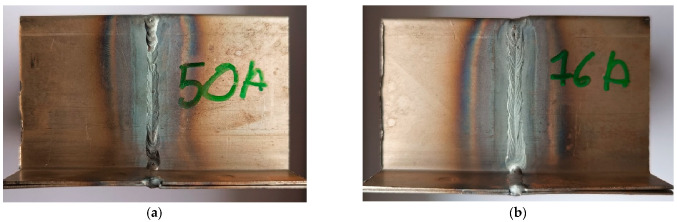
MAG welded parts: (**a**) specimen 5—welding current 50 A, (**b**) specimen 6—welding current 76 A.

**Figure 10 materials-18-03849-f010:**
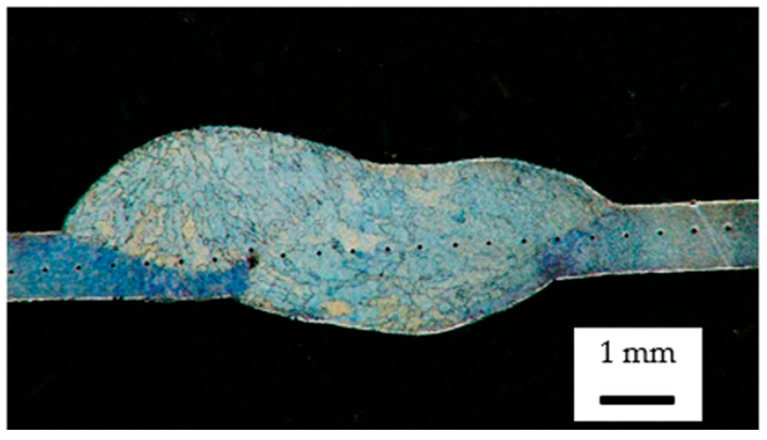
Hardness measurements on the cross-section of a welded joint—distribution of measurement points.

**Figure 11 materials-18-03849-f011:**
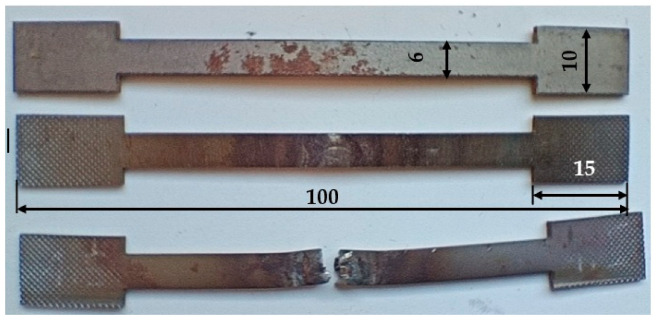
Mechanical property testing—sample specimens for tensile testing: source material, welded specimen, and specimen after rupture.

**Figure 12 materials-18-03849-f012:**
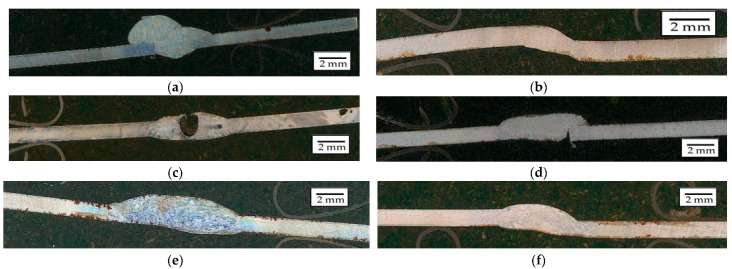
Examples of welding discrepancies: (**a**) excessive face convexity in specimen 4, (**b**) linear misalignment in specimen 5, (**c**) porosity in specimen 1, (**d**) lack of fusion in specimen 3 (**e**) without inconsistency—specimen 2, and (**f**) linear misalignment in specimen 6.

**Figure 13 materials-18-03849-f013:**
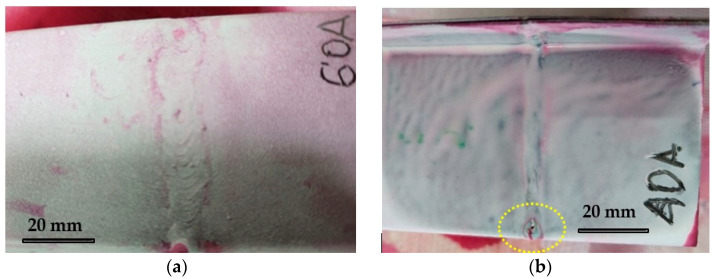
PT test results of MMA welds: (**a**) weld without irregularities—specimen 2, (**b**) weld irregularities - crater at the end of the weld—specimen 1.

**Figure 14 materials-18-03849-f014:**
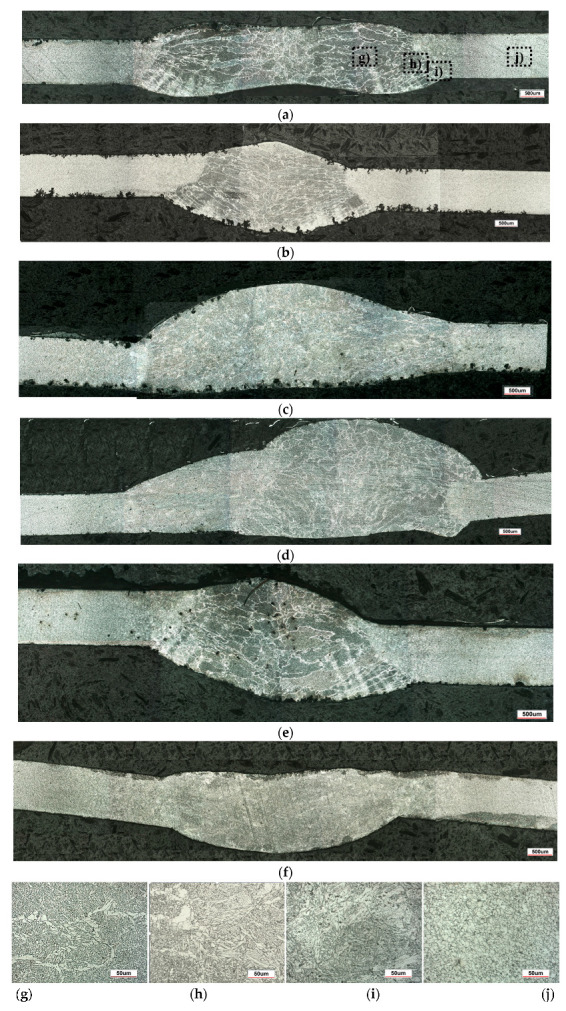
Structural changes in welded joint: (**a**) weld cross-section specimen 4 (magnification 50×), (**b**) weld cross-section specimen 1, (**c**) weld cross-section specimen 2, (**d**) weld cross-section specimen 3, (**e**) weld cross-section specimen 5, (**f**) weld cross-section specimen 6, (**g**) weld (fusion zone) (FZ), (**h**) weld + HAZ—fusion boundary, (**i**) HAZ, and (**j**) base material (BM), magnification 500×.

**Figure 15 materials-18-03849-f015:**
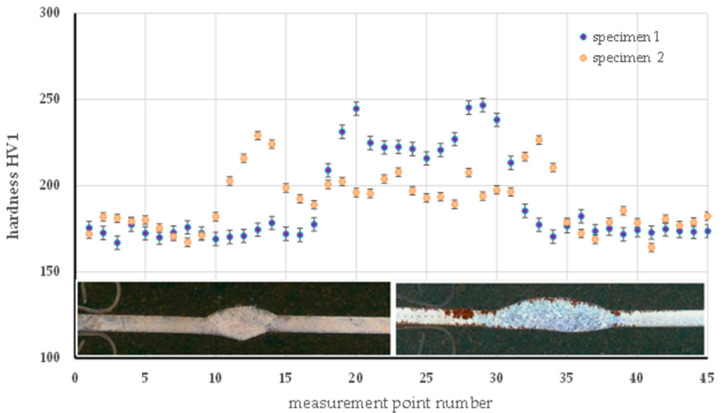
Hardness distribution on the cross-section of the specimen made by the MMA method—specimen 1 and specimen 2.

**Figure 16 materials-18-03849-f016:**
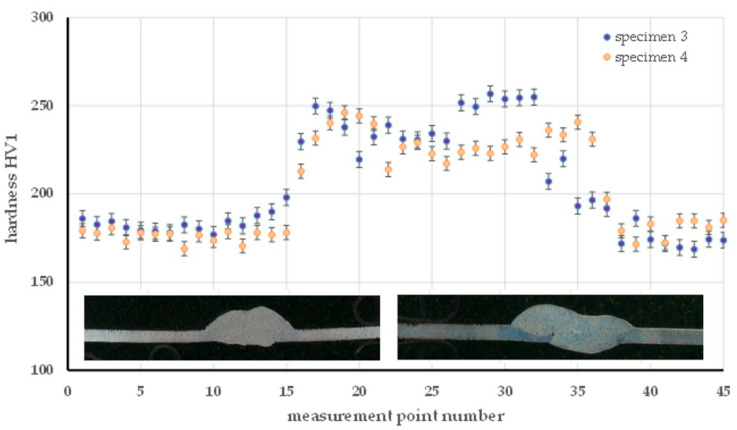
Hardness distribution on the cross-section of the specimen made by the MAG method—specimen 3 and specimen 4.

**Figure 17 materials-18-03849-f017:**
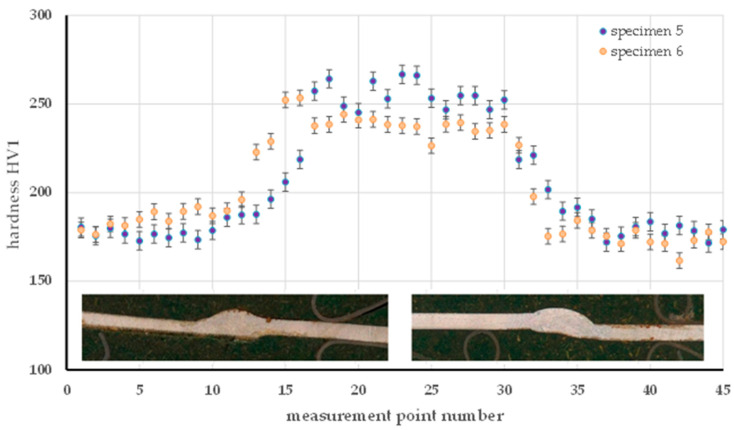
Hardness distribution on the cross-section of the specimen made by the TIG method—specimen 5 and specimen 6.

**Figure 18 materials-18-03849-f018:**
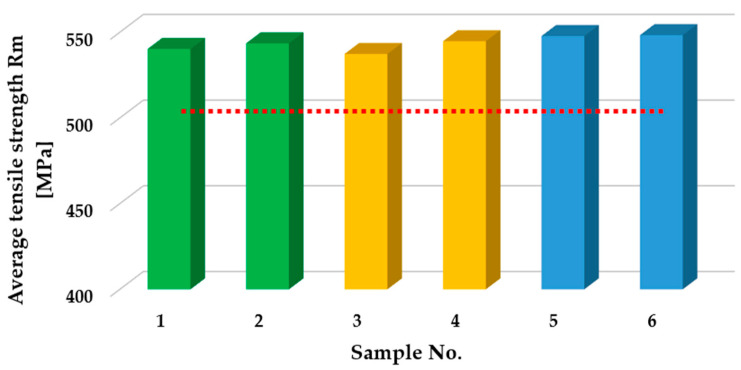
Comparison of the results of static tensile testing of HSLA 320 steel after welding and in the initial state. (MMA method - green color; MAG method - yellow color; TIG method - blue color; red line - the average tensile strength before welding)

**Table 1 materials-18-03849-t001:** Comparison of the chemical composition of HSLA 320 steel with the recommendations of the standard and measurements conducted in this study.

	Chemical Composition (wt %)					
	C	Mn	Si	Nb	Ti	V
Standard [27]	<0.21	0.40 to 1.50	0.05 to 0.50	<0.10		
Measurements	0.09	1.20	0.50	0.07	0.05	0.02

**Table 2 materials-18-03849-t002:** Mechanical properties of HSLA 320 steel according to the standard.

Re [MPa]	Rm [MPa]	A_80_ [%]	HV
≥320	420 to 520	≥20	130 to 190

**Table 3 materials-18-03849-t003:** Sample designation and welding parameters.

Sample Designation	Welding Method	Welding Current Intensity	Filler Wire	Shielding Gas
Specimen 1	MMA	40 A	alkaline electrode	none
Specimen 2		60 A		
Specimen 3	MAG	95 A	SG3 (1.2mm)	CO_2_ + Ar (mix)
Specimen 4		55 A		
Specimen 5	TIG	50 A	none	CO_2_ + Ar (mix)
Specimen 6		76 A		

## Data Availability

The data presented in this study are available on request from the corresponding author.

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
