# Peer review of "Evaluation of the Quality of Welded Joints After Repair of Automotive Frame Rails"

_materials, 2025, doi:10.3390/ma18163849_

Round 1

Reviewer 1 Report

Comments and Suggestions for Authors

The topic of reviewed manuscript is interesting. It is aimed at the current problem in the field of automotive materials.

The manuscript includes some deficiencies, which should be improved before publishing:

  • Table 1: it is not a table, but a figure. It should be revised,
  • Use in all decimal numbers decimal dot, not a decimal comma,
  • Section 5: write between section 5 and section 5.1 some “introducing text”, that a reader will now, what he can expect there,
  • Section 5.2: write between section 5.2 and 5.2.1 also some “introducing text”,
  • Figure 13, 14, 15: it includes too big letter font. Modify it, that font will be similar to the main text,
  • Figure 15: the manuscript includes two Figures No. 15,
  • The manuscript should be supplemented by the “Discussion”. It should include the description and conclusion of the research results, eventually compared with the existing research,
  • The literature should include newer and current research. There are included quite old sources,
  • I do not have major comment about the methodology. However, the comments above must be revised.

Author Response

Dear Reviewer,

thank you very much for your valuable comments and suggestions aimed at improving our manuscript. We greatly appreciate your,s feedback. We have revised the manuscript by addressing all applicable comments and suggestions. For any remarks that could not be incorporated directly, we have provided rebuttals. We hope that the current version of the manuscript is now significantly improved and more suitable for publication in Materials.

Please find enclosed the revised version of the manuscript (#materials-3763103), titled Evaluation of the Quality of Welded Joints After Repair of Automotive Frame Rails.

All changes in the manuscript are highlighted in green.

Reviewer's Notes

  • Table 1: it is not a table, but a figure. It should be revised,

Table 1 has been changed to Figure 3.

  • Use in all decimal numbers decimal dot, not a decimal comma,

Corrections have been made as suggested.

  • Section 5: write between section 5 and section 5.1 some “introducing text”, that a reader will now, what he can expect there,

Corrections have been implemented as recommended; currently at Chapter 6.

  • Section 5.2: write between section 5.2 and 5.2.1 also some “introducing text”,

Corrections have been implemented as recommended; currently at Section 6.2.

  • Figure 13, 14, 15: it includes too big letter font. Modify it, that font will be similar to the main text,

The fonts in the figures have been adjusted in accordance with the reviewers’ comments.

  • Figure 15: the manuscript includes two Figures No. 15,

Figure numbering has been revised accordingly.

  • The manuscript should be supplemented by the “Discussion”. It should include the description and conclusion of the research results, eventually compared with the existing research,

The discussion has been incorporated into both the Introduction and the Chapter 7.

  • The literature should include newer and current research. There are included quite old sources,

Several recent references have been added.

Reviewer 2 Report

Comments and Suggestions for Authors

The main reasons for rejecting the article are as follows:

Need for additional experiments – The experiments were not conducted correctly. Further characterization is required to strengthen the study.

Flawed research structure – The study lacks a proper chronological flow. A significant amount of information presented since section 2 does not align with the typical structure of a research article. 

Information was included in the attached file to improve the present article in a further submission.

Author Response

Dear Reviewer,

thank you very much for your valuable comments and suggestions aimed at improving our manuscript. We greatly appreciate your,s feedback. We have revised the manuscript by addressing all applicable comments and suggestions. For any remarks that could not be incorporated directly, we have provided rebuttals. We hope that the current version of the manuscript is now significantly improved and more suitable for publication in Materials.

Please find enclosed the revised version of the manuscript (#materials-3763103), titled Evaluation of the Quality of Welded Joints After Repair of Automotive Frame Rails.

Responses to comments from reviewer 2: (as per comments in the text). All corrections made in the text are highlighted in green. Figure numbering in the responses is consistent with the submitted (new) version of the manuscript.

  1. Choose a corresponding author.

Correction to the text

  1. Delete (HSLA 320).

Number 320 removed

  1. Show a scheme of where this part is located in a car.

Corected with an additional drawing: Fig 4. b) part located on the vehicle.

  1. Use impersonal redaction.

Corrected in text

  1. It is not clear what 0,40 ÷ 1,50 and 0,05 ÷ 0,5 mean. Clarify.

Corrected description in Table 1

  1. It is not clear what 420 ÷ 520 and 130 ÷ 190 mean. Clarify.

Corrected description in Table 2

  1. Add a general scheme of how the specimens were welded. And indicate the area of analysis where the specimens were obtained.

Completed with additional Figure 6 in the paper.

  1. To evaluate the efficiency of each weld method comparatively, the welding current intensity must be the same. It means, always use 40, 60 A, or 95, 55 A, or 50, 76 A. Justify why the welding parameters were selected, and support your response with citations.

The reviewer's remark is correct when comparing the suitability of a given method for welding the same element. The parameters in this case should be the same, but it is difficult to implement due to the specificity of each method. The aim of the reviewed work was not to compare welding methods, but to verify the correctness of welds in the repaired element of the vehicle. Therefore, the welds were commissioned to specialists who perform such repairs on a daily basis. In this case, the parameters were selected by them on the basis of practical knowledge and experience.

  1. Add the standard used.

Added item in text and literature [28]

  1. This is irrelevant.

Description of Fig. 10a removed from the text

  1. This is irrelevant.

Fig. 10a removed drawing and description

  1. Improve the scale bar; it is too small.

Added scale bare in  Fig. 10.

  1. Add dimensions; there are no dimensions included in the Figure 11.

Figure 11 corrected for specimen dimensions

  1. This is irrelevant.

Fig 11a and description removed from text.

  1. All the information presented in this section must be discussed and compared with specialized literature to increase the credibility of the results presented.

Chapter 7 discussion has been added and items 29-34 have been added to the literature.

  1. The six welded specimens must be shown to visualize the different welding inconsistencies and relate to the specific conditions of each test.

Two missing samples have been added (Fig. 12 e-f) and scales have been applied to the drawings.

  1. Add scale bars to both figures, and improve the mark of the welding defect in Figure 13b.

Scale bars were added to the drawing and the place of the identified non-compliance was marked in Fig. 13b

  1. The six welded specimens must be shown to visualize the different welding morphologies and relate to the specific conditions of each test. Also, add a scale bar to Figure 12c.

Added scale in fig. 12c. Figure 12 has been supplemented with a view of the remaining 5 samples (welds) – 12b-f. The figures show a view of the structure of the weld sample at 50x magnification. The differences in the welds are due to their shape, while the individual zones in HAZ are the same in all cases and are shown at 200x magnification in Fig. 12 g-j (generally for all cases).

  1. It is not clear what 280 ÷ 244 and 188 ÷ 229 mean. Clarify.

Correction of the text.

  1. It is necessary to show a phase characterization analysis to be able to suggest phase formations (ferrite and bainitic).

A description of the mechanism of formation of these phases during the welding process added in the text.

  1. It is not clear what 179 ÷ 182 means. Clarify. Skorygowano zapis w tekÅ›cie

Correction of the text

  1. Fig 15 (double numbering)

Updated the numbering of drawings at work

  1. Information about the manuscript is incomplete (Author Contributions, Funding, Consent statement, Data Availability Statement, and others). Please completed.

All information has been provided in the editorial system.

Reviewer 3 Report

Comments and Suggestions for Authors

The paper can be accepted, but it is necessary to make appropriate corrections in order to get it to the highest possible quality.

- The abstract, like the rest of the paper, contains a large number of abbreviations that are known but not explained in the text. At the first occurrence, explain each abbreviation and give its full name. Use either the full name or the abbreviation later in the text.

- The introductory part is unacceptably short for a journal of this rank. Expand the introductory part by analyzing papers from this field as well as applied methods. Explain what their advantages and disadvantages are. The introductory part of the paper should include a detailed analysis of previous research. In the mentioned literature, a large number of papers are older than 10 years, although the topic is quite current and there are a large number of new papers.

- Expand the Introductory Chapter with the following statement: "Advances in automotive technology indicate a trend toward faster, more powerful, and lighter vehicles." (Consult and add the following research: https://doi.org/10.46793/aeletters.2024.9.2.5. "Furthermore, the automotive sector is facing increasing pressure to improve design efficiency, reduce emissions, and adopt sustainable technologies, resulting in the need to redesign key vehicle components." (Consult and add the following research: https://doi.org/10.46793/adeletters.2025.4.2.3). 

- At the end of the introduction, what is the main contribution of the paper and how does it differ from similar ones in this field? What is the reason, i.e. why should this paper be published?

- In table 2, check whether it is the weight or volume content of individual elements.

- In line 159, instead of the dimension 300×60×60 mm, write 300mm×60mm×60 mm

- How many different samples were made for the same welding conditions and the same welding method? Provide a detailed explanation of why as many samples were made.

- Based on which recommendation were selected the parameters shown in table 4?

- How many times was the experiment repeated? Three or five times?

- Expand the discussion and analysis of the results. Compare the obtained results with the results of other researchers.

- Based on the extended analysis and discussion of the results, expand the concluding considerations.

Author Response

Dear Reviewer,

thank you very much for your valuable comments and suggestions aimed at improving our manuscript. We greatly appreciate your,s feedback. We have revised the manuscript by addressing all applicable comments and suggestions. For any remarks that could not be incorporated directly, we have provided rebuttals. We hope that the current version of the manuscript is now significantly improved and more suitable for publication in Materials.

Please find enclosed the revised version of the manuscript (#materials-3763103), titled Evaluation of the Quality of Welded Joints After Repair of Automotive Frame Rails.

All changes in the manuscript are highlighted in green.

Reviewer's Notes

- The abstract, like the rest of the paper, contains a large number of abbreviations that are known but not explained in the text. At the first occurrence, explain each abbreviation and give its full name. Use either the full name or the abbreviation later in the text.

Each abbreviation has been explained and provided with its full form.

- The introductory part is unacceptably short for a journal of this rank. Expand the introductory part by analyzing papers from this field as well as applied methods. Explain what their advantages and disadvantages are. The introductory part of the paper should include a detailed analysis of previous research. In the mentioned literature, a large number of papers are older than 10 years, although the topic is quite current and there are a large number of new papers.

The introductory section has been expanded and supplemented with additional discussion. Several recent references have also been added.

- Expand the Introductory Chapter with the following statement: "Advances in automotive technology indicate a trend toward faster, more powerful, and lighter vehicles." (Consult and add the following research: https://doi.org/10.46793/aeletters.2024.9.2.5. "Furthermore, the automotive sector is facing increasing pressure to improve design efficiency, reduce emissions, and adopt sustainable technologies, resulting in the need to redesign key vehicle components." (Consult and add the following research: https://doi.org/10.46793/adeletters.2025.4.2.3). 

Corrections have been implemented as recommended. The Introduction chapter has been expanded. The suggested literature items have been added and properly cited.

- At the end of the introduction, what is the main contribution of the paper and how does it differ from similar ones in this field? What is the reason, i.e. why should this paper be published?

The main contributions of the article are presented in the revised Introduction and elaborated upon in Chapter 2

- In table 2, check whether it is the weight or volume content of individual elements.

The table presents values as weight percentages and has been updated to include chemical composition data (wt%).

- In line 159, instead of the dimension 300×60×60 mm, write 300mm×60mm×60 mm

Corrections have been implemented as recommended

- How many different samples were made for the same welding conditions and the same welding method? Provide a detailed explanation of why as many samples were made.

A weld was executed around one stringer (four sides) for each welding method. The samples were prepared by two experienced welders, who independently selected the welding parameters based on their professional judgment.

- Based on which recommendation were selected the parameters shown in table 4?

The aim of the study was to assess the quality of welded joints in structural (load-bearing) vehicle components performed by unauthorized repair shops. Due to the absence of specific guidelines for welding parameters used in repairs at authorized service centers, the welders selected the parameters based on their own experience. While the chosen welding parameters appear adequate in terms of mechanical properties, they do not guarantee the restoration of the passive safety level originally specified by the manufacturer for the frame member.

- How many times was the experiment repeated? Three or five times?

Structural and hardness tests were conducted on individual metallographic specimens taken from randomly selected locations. Static tensile tests were carried out on five specimens cut from each of the six stringers. The results presented represent average values.

- Expand the discussion and analysis of the results. Compare the obtained results with the results of other researchers.

- Based on the extended analysis and discussion of the results, expand the concluding considerations.

The discussion has been incorporated into the Chapter 7, with appropriate references to relevant studies by other researchers.

Round 2

Reviewer 1 Report

Comments and Suggestions for Authors

The manuscript is well improved.
All my comments and remarks are included in the revised manuscript.
It can be published in the current form.

Author Response

Dear Reviewer,
thank you very much for your comments and suggestions aimed at improving our manuscript. 

Reviewer 2 Report

Comments and Suggestions for Authors

1.- Letters inside figure 14a must be renamed (g, h, i, j).

2.- In the final part, again, information about the manuscript is incomplete (Author Contributions, Funding, Consent statement, Data Availability Statement, and others). Check an example of a paper published on the web or use the template provided by the magazine. 

Author Response

Dear Reviewer,

thank you very much for your additional comments and suggestions aimed at improving our manuscript. We hope that the current version is now suitable for publication in Materials.Please find enclosed the revised manuscript (#materials-3763103), entitled Evaluation of the Quality of Welded Joints After Repair of Automotive Frame Rails.

All changes in the manuscript are highlighted in green.

Reviewer's Notes

1.- Letters inside figure 14a must be renamed (g, h, i, j).

Corrections have been implemented as recommended.

2.- In the final part, again, information about the manuscript is incomplete (Author Contributions, Funding, Consent statement, Data Availability Statement, and others). Check an example of a paper published on the web or use the template provided by the magazine.

Corrections have been implemented as recommended.

Reviewer 3 Report

Comments and Suggestions for Authors

The authors have answered all reviewer's comments. They have corrected and rewritten the applicable parts accordingly. Their explanations are complete and have been well addressed. I have no more comments to make. This review has improved the quality of the manuscript. For this reason, I suggest that the manuscript be accepted for publication.

Author Response

(The authors gave the same response as above.)
